# A multiplex chemiluminescent immunoassay for serological profiling of COVID-19-positive symptomatic and asymptomatic patients

Allison N. Grossberg[1,2,6], Lilia A. Koza [1,2,6], Aurélie Ledreux[1], Chad Prusmack[3], Hari Krishnan Krishnamurthy [4], Vasanth Jayaraman[4], Ann-Charlotte Granholm[1,7] & Daniel A. Linseman [1,2,5,7✉]

The COVID-19 pandemic affects more than 81 million people worldwide with over 1.7 million deaths. As the population returns to work, it is critical to develop tests that reliably detect SARS-CoV-2-specific antibodies. Here we present results from a multiplex serology test for assessing the antibody responses to COVID-19. In an initial large cohort, this test shows greater than 99% agreement with COVID-19 PCR test. In a second outpatient cohort consisting of adults and children in Colorado, the IgG responses are more robust in positive/symptomatic participants than in positive/asymptomatic participants, the IgM responses in symptomatic participants are transient and largely fall below the detection limit 30 days after symptom onset, and the levels of IgA against SARS-CoV-2 receptor binding domain are significantly increased in participants with moderate-to-severe symptoms compared to those with mild-to-moderate symptoms or asymptomatic individuals. Our results thus provide insight into serology profiling and the immune response to COVID-19.

[1] Knoebel Institute for Healthy Aging, University of Denver, Denver, CO, USA. [2] Department of Biological Sciences, University of Denver, Denver, CO, USA. [3] Resilience Code, Denver, CO, USA. [4] Vibrant Sciences LLC, San Carlos, CA, USA. [5] Eleanor Roosevelt Institute, University of Denver, Denver, CO, USA. [6] These authors contributed equally: Allison N. Grossberg, Lilia A. Koza. [7] These authors jointly supervised this work: Ann-Charlotte Granholm, Daniel A. Linseman. ✉email: daniel.linseman@du.edu

COVID-19 is an infectious disease caused by the novel coronavirus SARS-CoV-2[1–3]. Since December of 2019, COVID-19 has spread worldwide causing nearly 81 million infections and over 1.7 million deaths as of late-December of 2020 (https://www.who.int/emergencies/diseases/novel-coronavirus-2019/situation-reports, see also[2,4,5]). Novel COVID-19-infected pneumonia (NCIP) is characterized by fever, fatigue, dry cough, and dyspnea[3]. However, further study is required to accurately identify NCIP symptomology as a variety of other symptoms have been reported including vomiting, diarrhea, loss of smell, and loss of taste[1,6,7]. There is little information on symptom severity and presentation in COVID-19-positive patients and even less information on the antibody profile of asymptomatic patients infected with SARS-CoV-2. Most studies published since the first antibody tests were granted Emergency Use Authorization have focused on the detection of SARS-CoV-2 specific Immunoglobulin-M (IgM) and Immunoglobulin-G (IgG) antibodies against only one or two antigens (spike protein (SP) and/or nucleoprotein (NP))[8–12], and far fewer have analyzed the Immunoglobulin-A (IgA) antibody response in SARS-CoV-2 infected patients[13–15]. Therefore, additional research is needed to fully characterize the antibody response in asymptomatic and symptomatic SARS-CoV-2 infected patients.

Understanding the specific antibody profile and immune response in COVID-19 patients is critical for characterizing the progression of the disease, identifying patients with mild symptoms or who are asymptomatic or have delayed symptom onset, and predicting potential long-term immunity[16]. COVID-19-positive patients have undetectable antibody levels in the early stages of infection[17,18]. Liu et al. (2020) and others found that IgM antibodies were detected 4 days after symptom onset in COVID-19-positive patients and declined to undetectable levels after 4 weeks[16,18]. IgG antibodies reached detectable levels 7 days after infection, peaked around 25 days, and remained highly elevated after 4 weeks[18]. Another recent study of 173 COVID-19-positive patients showed that the median seroconversion times for IgM and IgG were 11 days and 14 days post-infection, respectively[17]. IgM levels began to decline after 5 weeks and were below detection limits after 7 weeks. IgG was highly elevated 7 weeks post-infection[17]. Given their long half-lives, IgG antibodies may remain above detectable thresholds for months or years after infection[19,20]. While some longitudinal studies have characterized the antibody response of COVID-19-positive patients and health-care workers[13], it is still unknown how long IgG may remain elevated after symptoms have subsided and what implications sustained IgG levels have on long-term immunological memory. Virus-specific IgM and IgA antibodies for SARS-CoV-1, which shares much of its genome with SARS-CoV-2[21], are similarly elevated in the serum of infected patients 6–8 days past symptom onset and begin to decline after 3–4 weeks[22,23]. Importantly, IgG titers in SARS-CoV-1 infected patients are elevated 8 weeks post-infection and remain elevated for up to 2 years which may indicate the presence of long-term immunity[24].

The most prominent immunoassays (IAs) are automated chemiluminescent IA, manual ELISA, and rapid lateral flow IA, which detect IgM and IgG produced in response to SARS-CoV-2 infection[8,25]. In the current manuscript, we describe a chemiluminescent immunoassay, developed by Vibrant America Clinical Labs, that is approved by the U.S. Food and Drug Administration (FDA) for Emergency Use Authorization and can sensitively detect antibody titers in SARS-CoV-2 infected patients. This serological protein microarray technology has been tested in more than 7,700 patient serum samples, and shown to detect a plurality of antibody responses including 12 antibody/antigen combinations: IgM, IgA, and IgG, against the spike 1 SP (S1 SP), receptor-binding domain (RBD), spike 2 SP (S2 SP), and NP of SARS-CoV-2[26]. Here, we present the overall clinical sensitivity and specificity for the Vibrant serology chemiluminescent immunoassay relative to PCR-confirmed COVID-19-positive and COVID-19-negative cases. We then utilize this seroanalytical tool to investigate the serology profile in a cohort of participants from Denver, Colorado.

Colorado has had over 326,000 cases of COVID-19, with 3717 deaths attributed to COVID-19, and 17,962 hospitalizations to date. Approximately 2.1 million individuals in the community have undergone testing (see https://covid19.colorado.gov/data/case-data for updates). Colorado is ranked number 21 in the country regarding the number of cases and number 26 in the country in terms of the number of reported deaths caused by COVID-19.

In the current study, we analyze symptomatology and antibody profiles of otherwise healthy, community-dwelling participants who reported a variety of COVID-like symptoms to an outpatient clinic in the Denver, Colorado area. Using the Vibrant multiplex chemiluminescent immunoassay, we find that the average IgG antibody titers against all SARS-CoV-2 antigens are significantly greater in COVID-19-positive symptomatic participants than in asymptomatic participants. COVID-19-positive symptomatic participants exhibit a greater IgG immune response while asymptomatic participants show a greater IgM response. Finally, elevated IgA levels track with increased symptom severity. These findings suggest that the Vibrant immunoassay may be useful in differentiating individual immune responses that are reflective of distinct symptom severities in those infected with SARS-CoV-2.

## Results

**Initial validation of Vibrant America Clinical Labs chemiluminescent immunoassay.** Although the Vibrant America Clinical Labs chemiluminescent immunoassay is recognized by the FDA for Emergency Use Authorization, this seroanalytical tool is relatively new[26]. Therefore, prior to using this chemiluminescent immunoassay to analyze the cohort of patients from Denver, Colorado, Vibrant performed a validation study which included serum samples from RT-PCR-confirmed COVID-19-positive ($n = 303$) and COVID-19-negative ($n = 296$) patients. In addition, serum samples from healthy controls ($n = 4502$) and non-COVID-19 disease controls ($n = 128$) were also used which were collected prior to the COVID-19 outbreak. Clinical sensitivity and specificity, as defined in the Methods, are displayed for IgA, IgG, and IgM antibodies against SARS-CoV-2 antigens S1, RBD, S2, and NP as assessed by the Vibrant America Clinical Labs chemiluminescent immunoassay (Table 1). The chemiluminescent immunoassay is sensitive enough to detect either IgA, IgG, or IgM for at least one antigen against SARS-CoV-2 in 99.67% of RT-PCR confirmed COVID-19-positive cases. On the other hand, the assay also demonstrates specificity in confirming that 99.77% of RT-PCR confirmed COVID-19-negative patients were negative for all antibody/antigen combinations (Table 1). These data indicate that the Vibrant America Clinical Labs chemiluminescent immunoassay demonstrates high clinical sensitivity and specificity and can reliably be used to diagnose infection by SARS-CoV-2.

**Participant population – Denver, Colorado cohort.** Serum samples from 107 participants (52 males and 55 females) between the ages of 12 and 78 (mean = 42 years) living in Denver, Colorado and the surrounding suburbs, were tested for the presence of IgA, IgG, and IgM antibodies against SARS-CoV-2 antigens, including S1, RBD, S2, and NP using the Vibrant America Clinical Labs chemiluminescent immunoassay[26]. Thirty-one participants tested positive for at least one SARS-CoV-2

**Table 1 Vibrant America Clinical Labs automated chemiluminescent immunoassay validation.**

| Antibody/Antigen | Clinical Sensitivity (95% Confidence) | Clinical Specificity (95% Confidence) |
|---|---|---|
| Overall IgM/IgG/IgA | 99.67% (98.15–99.94%) | 99.77% (99.60–99.87%) |
| Overall IgM | 87.46% (83.25–90.73%) | 99.81% (99.65–99.90%) |
| Overall IgG | 99.01% (97.13–99.66%) | 99.77% (99.60–99.87%) |
| Overall IgA | 61.72% (56.13–67.01%) | 99.90% (99.78–99.96%) |
| S1 SP IgM | 70.30% (64.92–75.16%) | 99.85% (99.70–99.92%) |
| S1 SP IgG | 77.23% (72.18–81.59%) | 99.85% (99.70–99.92%) |
| S1 SP IgA | 52.15% (46.53–57.71%) | 99.92% (99.80–99.97%) |
| RBD IgM | 66.34% (60.84–71.42%) | 99.83% (99.68–99.91%) |
| RBD IgG | 66.34% (60.84–71.42%) | 99.79% (99.63–99.88%) |
| RBD IgA | 41.25% (35.85–46.87%) | 99.90% (99.78–99.96%) |
| S2 SP IgM | 67.99% (62.54–72.99%) | 99.81% (99.65–99.90%) |
| S2 SP IgG | 81.52% (76.76–85.48%) | 99.83% (99.68–99.91%) |
| S2 SP IgA | 41.58% (36.17–47.21%) | 99.90% (99.78–99.96%) |
| NP IgM | 63.37% (57.81–68.59%) | 99.81% (99.65–99.90%) |
| NP IgG | 70.96% (65.61–75.78%) | 99.77% (99.60–99.87%) |
| NP IgA | 32.34% (27.33–37.80%) | 99.94% (99.83–99.98%) |

Serum samples from RT-PCR confirmed COVID-19-positive ($n = 303$) and COVID-19-negative ($n = 296$) patients, along with healthy controls ($n = 4,502$) and non-COVID-19 disease controls ($n = 128$) collected prior to the COVID-19 outbreak, were analyzed by the Vibrant America Clinical Labs chemiluminescent immunoassay. Clinical sensitivity and specificity are displayed for IgM, IgG, and IgA antibodies against SARS-CoV-2 spike 1 glycoprotein (S1 SP), spike 2 glycoprotein (S2 SP), receptor-binding domain (RBD), and nucleoprotein (NP). Clinical sensitivity and specificity for overall IgM, IgG, IgA, and for overall IgM/IgG/IgA are also displayed. Data are presented as % (95% confidence levels). A more comprehensive analysis of the sensitivity and specificity of the Vibrant chemiluminescent immunoassay for SARS-CoV-2 has recently been published[26].

antibody/antigen combination (antibody titers ≥ 1.0). Seventy-six participants tested negative for all antibody/antigen combinations (titers < 1.0). Sixty-four of the participants (28 COVID-19-positive and 36 COVID-19-negative) completed extensive health questionnaires to identify symptom severity and potential comorbidities. Serological testing was performed at one timepoint and participants who reported symptoms were tested an average of 50 days (range 1–117 days) relative to symptom onset. Twenty-one COVID-19-positive participants and 18 COVID-19-negative participants reported experiencing one or more symptoms typically associated with COVID-19. As assessed by patient responses to questionnaire items related to medical history, physical activity levels, and general lifestyle practices, this population was considered to be healthy and active. This population had an average body mass index (BMI) of 25, only 21 participants reported chronic illnesses, and only 10 participants had a history of smoking tobacco or marijuana (Table 2). There were no significant differences in age, sex, BMI, smoking history, or the presence of chronic illnesses between the COVID-19-positive and COVID-19-negative participants (Table 2).

**Antibody titers against SARS-CoV-2 antigens.** We first confirmed that all combinations of antibody titers against SARS-CoV-2 antigens were significantly elevated in COVID-19-positive participants ($n = 31$) versus COVID-19-negative participants ($n = 71$) using two-tailed Independent Samples t-Tests. Titers for all 12 antibody/antigen combinations were significantly increased in COVID-19-positive versus COVID-19-negative participants (Supplementary Fig. 1). Titers were not significantly different between males and females (Supplementary Fig. 2).

**Loss of smell was uniquely more severe in COVID-19-positive participants.** Several participants in both the COVID-19-positive and COVID-19-negative groups reported symptoms. Therefore, we analyzed differences in symptom presentation and severity between the 21 symptomatic participants that tested positive and the 18 symptomatic participants that tested negative using two-tailed Independent Samples T-Tests. All participants self-reported the first date they remembered experiencing symptoms, symptom severity, and the date on which the symptoms resolved. Days between symptom onset and symptom resolution were not

significantly different between the COVID-19-positive symptomatic participants and COVID-19-negative symptomatic participants (Supplementary Table 1). Eighteen different symptoms were reported with varying severities between the symptomatic COVID-19-positive and COVID-19-negative participants. Severity of loss of smell was the only symptom significantly increased in positive participants (mean(SE) = $4.50 \pm 0.50$) when compared to negative participants (mean(SE) = $2.50 \pm 0.65$) ($t(6) = -2.45$, $p = 0.05$, Supplementary Table 1). These data suggest that severe loss of smell may be uniquely associated with SARS-CoV-2 infection. We ran Pearson's correlations with the total symptom severity scores for each symptom and the compiled total score and found that none of the 18 symptoms were significantly correlated with age or BMI and there were no significant differences in symptom severity between males or females for any of the reported symptoms.

**Antibody titers against SARS-CoV-2 antigens in participants with mild versus severe symptoms.** Although no participants in the present study required hospitalization, both COVID-19-negative and COVID-19-positive participants reported experiencing symptoms that were severe (defined as a symptom that could not be ignored and was the worst the patient had ever felt) or moderate (defined as a symptom that could not be ignored and limited daily activities). Since not all COVID-19-positive participants tested positive for every antibody/antigen combination, we investigated whether COVID-19-positive participants with more severe symptomology were more likely to be positive for specific antibody/antigen combinations. We compared antibody titers against SARS-CoV-2 antigens in COVID-19-positive participants who reported having mild-moderate symptoms ($n = 11$) to those who reported moderate-severe symptoms ($n = 10$) and to those who were asymptomatic ($n = 7$). Titers of IgA antibodies against RBD were statistically significantly higher in COVID-19-positive participants with moderate-severe symptoms (mean(SE) = $0.93 \pm 0.19$) when compared to participants with mild-moderate symptoms (mean(SE) = $0.45 \pm 0.05$) and asymptomatic participants (mean(SE) = $0.41 \pm 0.10$) ($F(2,25) = 5.73$, $p = 0.01$, Fig. 1a). We also found that titers of IgG against NP as well as IgA against S2 SP were significantly elevated in participants with moderate-severe symptoms (IgG NP: (mean

**Table 2 Demographic and clinical characteristics of the Colorado cohort including COVID-19-positive and COVID-19-negative participants.**

| | Total study population | COVID-19-positive | COVID-19-negative | *p*-Value |
|---|---|---|---|---|
| **Total sample size (*n*)** | 107 | 31 | 76 | — |
| **Age, years** | 42.2 (1.53) [12–78] | 40.7 (2.75) [12–68] | 42.7 (1.84) [12–78] | *p* = 0.56 |
| **Male sex** | 52 (48.6%) | 12 (38.7%) | 40 (52.6%) | *p* = 0.19 |
| **Questionnaire Sample Size (*n*)** | 64 | 28 | 36 | — |
| **Days between symptom onset and serological test** | 49.8 (5.13) [1–117] | 44.4 (6.68) [6–117] | 56.4 (7.89) [1–114] | *p* = 0.25 |
| **Ethnicity** | White (61) (95.3%) Asian (2) (3.1%) Hispanic, Latino or Spanish (1) (1.6%) | White (25) (89.3%) Asian (2) (7.1%) Hispanic, Latino or Spanish (1) (3.6%) | White (36) (100.0%) | *p* = 0.08 |
| **BMI** | 25.2 (0.56) [18.6–43.3] | 25.2 (0.84) [20.2–43.3] | 25.2 (0.75) [18.6–41.1] | *p* = 0.97 |
| **Smoking history (tobacco or marijuana)** | 10 (15.6%) | 4 (14.3%) | 6 (16.7%) | *p* = 1.00 |
| **Hypertension** | 0 (0.0%) | 0 (0.0%) | 0 (0.0%) | — |
| **Cardiovascular Disease** | 4 (6.3%) | 0 (0.0%) | 4 (11.1%) | *p* = 0.12 |
| **Diabetes** | 1 (1.6%) | 1 (3.6%) | 0 (0.0%) | *p* = 0.44 |
| **Liver disease** | 0 (0.0%) | 0 (0.0%) | 0 (0.0%) | — |
| **Autoimmune Disease** | 5 (7.8%) | 2 (7.1%) | 3 (8.3%) | *p* = 1.00 |
| **Tick-Borne Illness** | 7 (10.9%) | 2 (7.1%) | 5 (13.9%) | *p* = 0.45 |
| **Hemochromatosis** | 1 (1.6%) | 1 (3.6%) | 0 (0.0%) | *p* = 0.44 |
| **Blood related disease (i.e., anemia)** | 3 (4.7%) | 2 (7.1%) | 1 (2.8%) | *p* = 0.58 |

Population percentages for ethnicity, smoking history (tobacco or marijuana), hypertension, cardiovascular disease, diabetes, liver disease, autoimmune disease, tick borne illness, hemochromatosis, blood-related disease (i.e., anemia), and days between symptom onset and serological testing were calculated based on the number of total (*n* = 64), COVID-19-positive (*n* = 28), and COVID-19-negative (*n* = 36) participants who responded to the questionnaire (shaded section of Table). Data are presented as the mean (SE), [range], or number (%). Statistical differences are shown for COVID-19-positive versus COVID-19-negative participants. *p*-values were determined via two-tailed Independent Samples T-Tests.

(SE) = 1.10 ± 0.12); IgA S2 SP: (mean(SE) = 0.80 ± 0.10)) compared to those who were asymptomatic (IgG NP: (mean(SE) = 0.54 ± 0.10), $t(15) = -2.36$, $p = 0.03$; IgA S2 SP: (mean(SE) = 0.50 ± 0.10), $t(15) = -2.24$, $p = 0.04$; Fig. 1b, c). We did not find significant differences in titers of IgA antibodies against RBD, IgA against S2 SP, or IgG against NP between COVID-19-positive female and male participants in these symptom groups (Supplementary Fig. 3). There were also no significant differences in days between symptom onset to serological test date, age, sex, and BMI between COVID-19-positive participants with mild-moderate versus moderate-severe symptomology.

**IgM and IgG timelines.** Next, we assessed the variation between reported symptom onset and serology test date. There were no significant correlations between titers and days between symptom onset and resolution or days between symptom onset and the initial test. However, within the COVID-19-positive population, subgroups of participants with high titers of IgG and IgM antibodies stratified to distinct intervals after symptom onset (Fig. 2a, b). COVID-19-positive participants displayed elevated levels of IgM and IgG (each immunoglobulin averaged for the four antigens tested) between 0–30 and 0–60 days, respectively, after symptom onset. IgM levels appeared to decline 30 days past symptom onset.

Based on the findings described above, we next compared all combinations of antibody titers against SARS-CoV-2 antigens in the 21 symptomatic COVID-19-positive participants at 0–30, 30–60, 60–90, and 90–120 days between symptom onset and the serology test date using one-way ANOVA tests. Average IgG antibody concentration was not significantly different between each interval but was higher between 0 and 30 days (mean(SE) = 1.14 ± 0.18) compared to 30-60 days (mean(SE) = 1.03 ± 0.23, Fig. 2a). Average IgM titers were significantly different between the 0–30 and 30–60 day intervals ($p = 0.03$) and between the 0–30 and 60–90 day intervals ($p = 0.05$) between symptom onset and the initial test ($F(3,17) = 4.47$, $p = 0.02$, Fig. 2b). The

0–30 day interval had the highest average IgM concentration (mean(SE) = 1.08 ± 0.11, $p = 0.03$) compared to the other intervals. The average levels of IgM antibodies against the S1 ($F(3,17) = 10.89$, $p < 0.01$; Fig. 2c) and RBD ($F(3,17) = 4.55$, $p = 0.02$; Fig. 2d) antigens were significantly different between the four intervals described above. Specifically, there were significant differences between the 0–30 ($p < 0.01$) and 30–60 ($p = 0.04$) day intervals, as well as the 0–30 ($p < 0.01$) and 60–90 ($p = 0.04$) day intervals for the S1 antigen and the RBD, respectively (Fig. 2c, d).

**Antibody titers in COVID-19-positive symptomatic versus asymptomatic participants.** Differences in demographics and clinical characteristics between asymptomatic and symptomatic COVID-19-positive participants were examined next. There were no significant differences in sex, age, BMI, smoking history, or the presence of chronic illnesses between these two groups when analyzed by two-tailed Independent Samples t-Tests (Table 3). Only titers for IgG against RBD were significantly increased in symptomatic females ($n = 6$) versus symptomatic males ($n = 4$, $t(8) = 2.71$, $p = 0.03$). Fisher's Exact Tests were used to compare the proportion of asymptomatic and symptomatic participants who tested positive for each antibody/antigen combination. We found that a greater number of symptomatic participants tested positive for IgG against the S1 antigen ($n = 10$, $p = 0.03$) and the NP antigen ($n = 10$, $p = 0.03$) when compared to asymptomatic COVID-19-positive participants ($n = 0$ for both S1 and NP; Table 3). Similarly, more symptomatic participants tested positive for IgG against S2 ($n = 12$) than asymptomatic participants ($n = 1$), and this result trended towards significance ($p = 0.08$, Table 3). The titers for the remaining antibody/antigen combinations were compared via two-tailed Independent Samples T-Tests and there were no significant differences between the two groups (Table 3). Finally, we utilized average titers for IgA, IgG, and IgM against all antigens and examined differences in COVID-19-positive asymptomatic and symptomatic participants using two-tailed Independent Samples T-Tests. We found that the

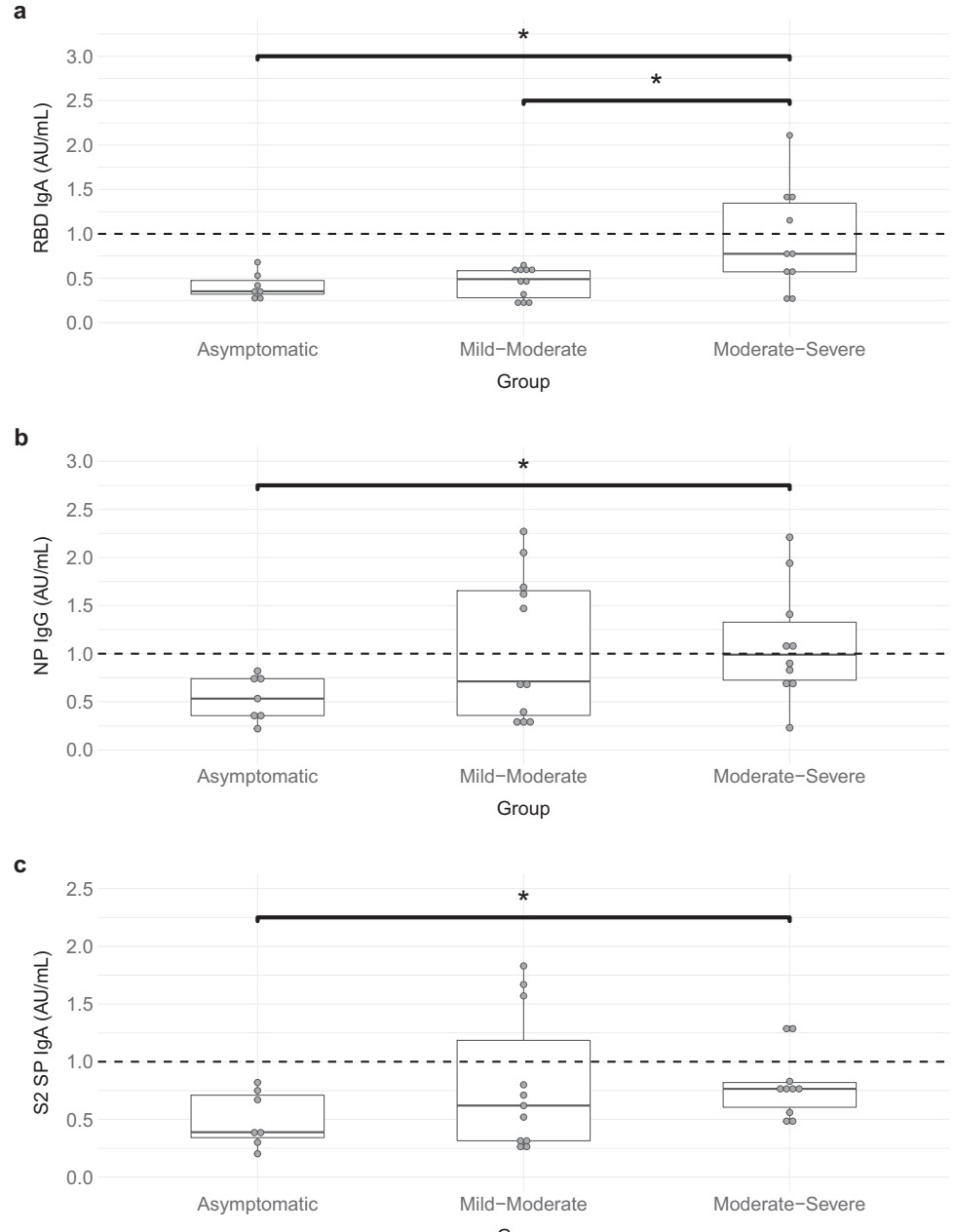

**Fig. 1 IgA antibody titers against receptor-binding domain (RBD) and spike 2 glycoprotein (S2 SP) and IgG antibody titers against nucleoprotein (NP) plotted against symptom severity in COVID-19-positive participants.** Comparison of **a** IgA antibody titers against RBD, **b** IgG antibody titers against nucleoprotein, and **c** IgA antibody titers against S2 SP in COVID-19-positive participants with no symptoms (asymptomatic, $n = 7$), mild-moderate symptom severity ($n = 11$), and moderate-severe symptom severity ($n = 10$). Boxplots represent the minimum, maximum, median, first quartile and third quartile in the data set. The cut-off value for the serological test ($\geq 1$) is shown as a black dashed line for reference. IgA antibody titers against RBD were analyzed using one-way ANOVA and Tukey post hoc tests were used for multiple comparisons. IgA RBD titers were significantly higher in participants with moderate-severe symptoms compared to participants with mild-moderate symptoms ($p = 0.02$) and asymptomatic participants ($p = 0.02$). IgA antibody titers against S2 SP and IgG antibody titers against nucleoprotein were analyzed using two-tailed Independent Samples T-tests. IgG NP and IgA S2 SP titers were significantly elevated in participants with moderate-severe symptoms compared to those who were asymptomatic (IgG NP: $p = 0.03$; IgA S2 SP: $p = 0.04$). * indicates $p < 0.05$.

average IgG antibody titers against all antigens showed a trend towards being greater in COVID-19-positive symptomatic participants (mean(SE) = 1.04 ± 0.11) than COVID-19-positive asymptomatic participants (mean(SE) = 0.64 ± 0.10), ($t(26) = -2.00$, $p = 0.06$). These data suggest that asymptomatic and symptomatic COVID-19-positive participants may experience different immune responses to SARS-CoV-2 infection. We utilized a Fisher's Exact Test to explore associations between IgA,

IgG, or IgM antibody positivity and the presence or absence of symptoms. We found that a larger number of asymptomatic COVID-19-positive participants had at least one positive IgM antigen compared to the number with at least one positive IgG antigen (Fisher's Exact test, $p = 0.05$). Collectively, the above findings suggest that COVID-19-positive symptomatic participants exhibited a greater IgG immune response and that asymptomatic participants had a greater IgM response.

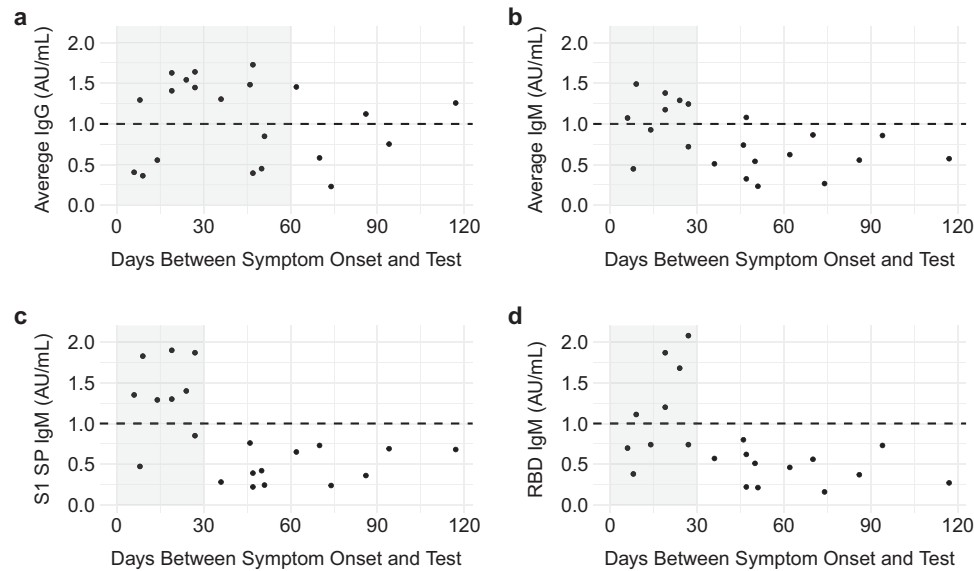

**Fig. 2 IgG and IgM antibody titers plotted against days between symptom onset and serological test for COVID-19-positive participants.** Black dots represent **a** average IgG antibody titers, **b** average IgM antibody titers, **c** average IgM antibody titers against the spike 1 glycoprotein (S1 SP), and **d** average IgM antibody titers against the receptor-binding domain (RBD) in symptomatic COVID-19-positive ($n = 21$) participants plotted against days between symptom onset and serological test. The cut-off value for the serological test ($\geq 1$) is shown as a black dashed line for reference. Shaded areas represent the time interval during which the majority of black dots are in the positive range (>=1). All data were analyzed using one-way ANOVA. Tukey post hoc tests were used for multiple comparisons.

**Table 3 Comparison of IgM, IgA, and IgG titers against SARS-CoV-2 spike 1 glycoprotein (S1 SP), spike 2 glycoprotein (S2 SP), receptor-binding domain (RBD), and nucleoprotein (NP) in symptomatic and asymptomatic COVID-19-positive participants.**

|  | COVID-19-Positive: Symptomatic | COVID-19-Positive: Asymptomatic | *p*-Value |
|---|---|---|---|
| **Sample size (*n*)** | 21 | 7 |  |
| **Age** | 38.5 (3.25) [12–62] | 37.7 (4.30) [26–53] | $p = 0.90$ |
| **Male sex** | 8 (38.1%) | 1 (14.3%) | $p = 0.37$ |
| **BMI** | 25.4 (1.11) [20.2–43.3] | 24.6 (0.68) [21.8–27.1] | $p = 0.67$ |
| **S1 SP IgM** | 7 (33.3%) | 3 (42.9%) | $p = 0.67$ |
|  | 1.56 (0.11) [1.29–1.90] | 1.35 (0.16) [1.06–1.59] | $p = 0.32$ |
| **S1 SP IgG** | 10 (47.6%) | 0 (0.0%) | $p = 0.03$ |
|  | 1.58 (0.13) [1.09–2.24] | — | — |
| **S1 SP IgA** | 2 (9.5%) | 0 (0.0%) | $p = 1.00$ |
|  | 1.43 (0.39) [1.05–1.82] | — | — |
| **RBD IgM** | 5 (23.8%) | 2 (28.6%) | $p = 1.00$ |
|  | 1.59 (0.19) [1.11–2.08] | 1.31 (0.02) [1.29–1.33] | $p = 0.42$ |
| **RBD IgG** | 10 (47.6%) | 1 (14.3%) | $p = 0.19$ |
|  | 1.57 (0.10) [1.17–2.14] | 1.92 (NA) [NA] | $p = 0.32$ |
| **RBD IgA** | 4 (19.0%) | 0 (0.0%) | $p = 0.55$ |
|  | 1.52 (0.21) [1.15–2.11] | — | — |
| **S2 SP IgM** | 7 (33.3%) | 3 (42.9%) | $p = 0.67$ |
|  | 1.43 (0.12) [1.08–1.99] | 1.39 (0.05) [1.29–1.47] | $p = 0.82$ |
| **S2 SP IgG** | 12 (57.1%) | 1 (14.3%) | $p = 0.08$ |
|  | 1.57 (0.12) [1.05–2.21] | 2.04 (NA) [NA] | $p = 0.31$ |
| **S2 SP IgA** | 5 (23.8%) | 0 (0.0%) | $p = 0.30$ |
|  | 1.53 (0.11) [1.28–1.83] | — | — |
| **NP IgM** | 5 (23.8%) | 4 (57.1%) | $p = 0.17$ |
|  | 1.51 (0.13) [1.04–1.79] | 1.45 (0.21) [1.18–2.07] | $p = 0.83$ |
| **NP IgG** | 10 (47.6%) | 0 (0.0%) | $p = 0.03$ |
|  | 1.68 (0.14) [1.08–2.27] | — | — |
| **NP IgA** | 3 (14.3%) | 0 (0.0%) | $p = 0.55$ |
|  | 2.40 (0.82) [1.24–3.99] | — | — |

Age, sex, and BMI are also shown for comparison in symptomatic ($n = 21$) and asymptomatic ($n = 7$) COVID-19-positive participants. For each antibody/antigen combination, the number of positive participants (%) is indicated. Titer values are given in AU/mL and results are presented as mean (SE) [range]. *p*-values were determined via two-tailed Independent Samples t-Tests for continuous data, Chi-Squared Tests for categorical data with >5 cases per cell, and Fisher's Exact Tests for categorical data with <5 cases per cell.

Intriguingly, none of the COVID-19-positive asymptomatic participants were positive for IgA against any viral antigen tested (Table 3).

**Antibody/antigen combinations in relation to COVID-19 diagnosis and symptom severity.** To better understand whether certain antibody/antigen combinations were important in establishing a COVID-19-positive diagnosis, and whether certain antibody/antigen combinations were characteristic of symptom severities, we used Pearson correlations. We included COVID-19-positive ($n = 31$) and negative participants ($n = 76$, Supplementary Fig. 4a) and separated COVID-19-positive participants with no symptoms ($n = 7$, asymptomatic), mild-moderate symptoms ($n = 11$), and moderate-severe symptoms ($n = 10$, Supplementary Fig. 4b). In COVID-19-positive participants, we found that IgG against S1 SP and IgG against RBD ($r = 0.7$), as well as IgM against S1 SP and IgM against RBD ($r = 0.7$) were strongly positively correlated (Supplementary Fig. 4a). No strong positive or negative correlations were seen between the antibody/antigen combinations in COVID-19-negative participants (Supplementary Fig. 4a). Furthermore, some positive correlations were observed when COVID-19-positive participants were further subdivided by symptom severity, although there were too few participants in each group to draw any definitive conclusions (Supplementary Fig. 4b). Intriguingly, IgM antibodies against S1 SP and IgM antibodies against RBD were positively correlated in participants with mild-moderate symptoms ($r = 0.9$) and with moderate-severe symptoms ($r = 0.9$) but not in asymptomatic participants ($r = 0.5$, Supplementary Fig. 4b). However, further study is required as these correlational data are difficult to interpret due to relatively small sample sizes.

Finally, titers from participants with varying symptomology were individually plotted to visualize differences in antibody profiles and to explore whether increased symptom severity resulted in an increase in the magnitude of the antibody response, indicative of increased titers for many antibody/antigen combinations. A heatmap was used to display increased and decreased titers for each antibody/antigen combination for individual COVID-19-positive participants experiencing no symptoms (asymptomatic, $n = 7$), participants experiencing mild-moderate symptoms ($n = 10$), and participants with moderate-severe symptoms ($n = 10$) (Fig. 3). Overall, it does appear that most COVID-19-positive asymptomatic participants had increased titers ($\geq 1.0$) for only a few antibody/antigen combinations. In contrast, many participants experiencing moderate-severe symptoms seemed to have increased titers for several antibody/antigen combinations while participants with mild-moderate symptoms had increased titers for fewer antibody/antigen combinations (Fig. 3).

Consistent with our findings reported above, we also observed that symptomatic participants displayed increased titers for IgG antibodies, while asymptomatic participants showed increased titers for IgM antibodies against most antigens. We also see an increase in titers for IgA antibodies against RBD in participants with moderate-severe symptoms compared to asymptomatic and participants with mild-moderate symptoms (Fig. 3). Taken together, these data may provide insight into how the magnitude of the antibody response potentially influences symptom severity.

## Discussion

In the current study, we utilized a multiplex serology test for COVID-19 in an otherwise healthy cohort of adults and children in Colorado. Recent results using the same test kit have shown that IgM antibodies against SARS-CoV-2 are generally detectable in blood several days after initial infection, although levels over time are not well characterized[26]. Our results show that IgM levels and IgG levels were both elevated early (0–30 days following symptom onset). IgM levels declined 30 days following symptom onset while IgG levels remained elevated for up to 60 days following symptom onset. Increased early IgM levels may indicate acute infection and a later elevation in IgG levels correspond to a prolonged immune response and the activation of adaptive humoral immunity[8]. Suhandynata and collaborators (2020)[27] demonstrated the evolution of seroconversion for both IgG and IgM in a cohort of acutely ill patients, supporting our findings here. Although IgG levels declined after 60 days post-symptom onset, a recent study indicates that relatively low levels of IgG may still provide significant immunity against SARS-CoV-2[28].

Contrary to these results, several studies have shown that IgM levels peaked around day 20 and fell below baseline by about 6 months, while IgG levels remained elevated beyond 2 years post-infection in SARS-CoV-1 infected patients[22,24]. Despite the similarities between SARS-CoV-1 and SARS-CoV-2, it is possible that IgG levels do not remain elevated long-term for COVID-19 patients. Recently, several other studies have reported that IgG levels against SARS-CoV-2 were still elevated in COVID-19-positive patients after 50 days[29]. Similar results were described by Zhang et al. (2020), but there is little research on the long-term titers of IgG in a larger cohort[25]. Recurrence of COVID-19 has not been well described, suggesting that antibody presence could confer at least short-term immunity. This is in agreement with the findings of Chandrashekar et al. (2020) which showed that primates infected with SARS-CoV-2 were protected from reinfection following viral clearance[30]. Interestingly, IgG antibody development against SARS-CoV-2 has also been associated with a reduced viral load in the respiratory tract[31]. These findings may indicate antibody development offers some level of protection from reinfection. Furthermore, it is generally accepted that patients with severe illness are more likely to have heightened immune responses and robust long-term immunity than those with milder illness[32]. Thus, it should be noted that the cohort studied here represents a relatively young, healthy population which did not require hospitalization. It is possible that the decline in IgG levels may be due to a less robust immune response to a relatively mild infection[12,19,33]. Serology profiles may look different in an elderly population, or those that have been primed by previous or chronic infectious diseases or conditions.

The current study contained a comprehensive symptom inventory, including respiratory symptoms, neurological symptoms, and GI-related symptoms (see Supplementary Table 1). When comparing symptomology between COVID-19-positive and COVID-19-negative participants, we hoped to identify symptoms that were uniquely associated with COVID-19 infection and that correlated with immune profiles. Levels of most antibody/antigen combinations were elevated in COVID-19-positive participants with moderate-severe symptoms when compared to asymptomatic participants or those with mild-moderate symptoms. The only symptom unique to positive participants was loss of smell. In a study by Dell'Era et al. (2020), investigators examined the loss of taste and smell in a cohort of Italian patients with confirmed COVID-19[34]. They found that 70% of this population experienced loss of taste or smell. In line with our findings, Menni et al. (2020) showed that out of 10 commonly reported symptoms of COVID-19, loss of smell and taste were the best predictors of a positive test result[35]. It has been reported that loss of taste and smell may be the earliest signs of infection and thus, may represent a marker of viral shedding and a reliable predictor of infection.

We found that the average IgG antibody titers against all antigens were significantly greater in COVID-19-positive

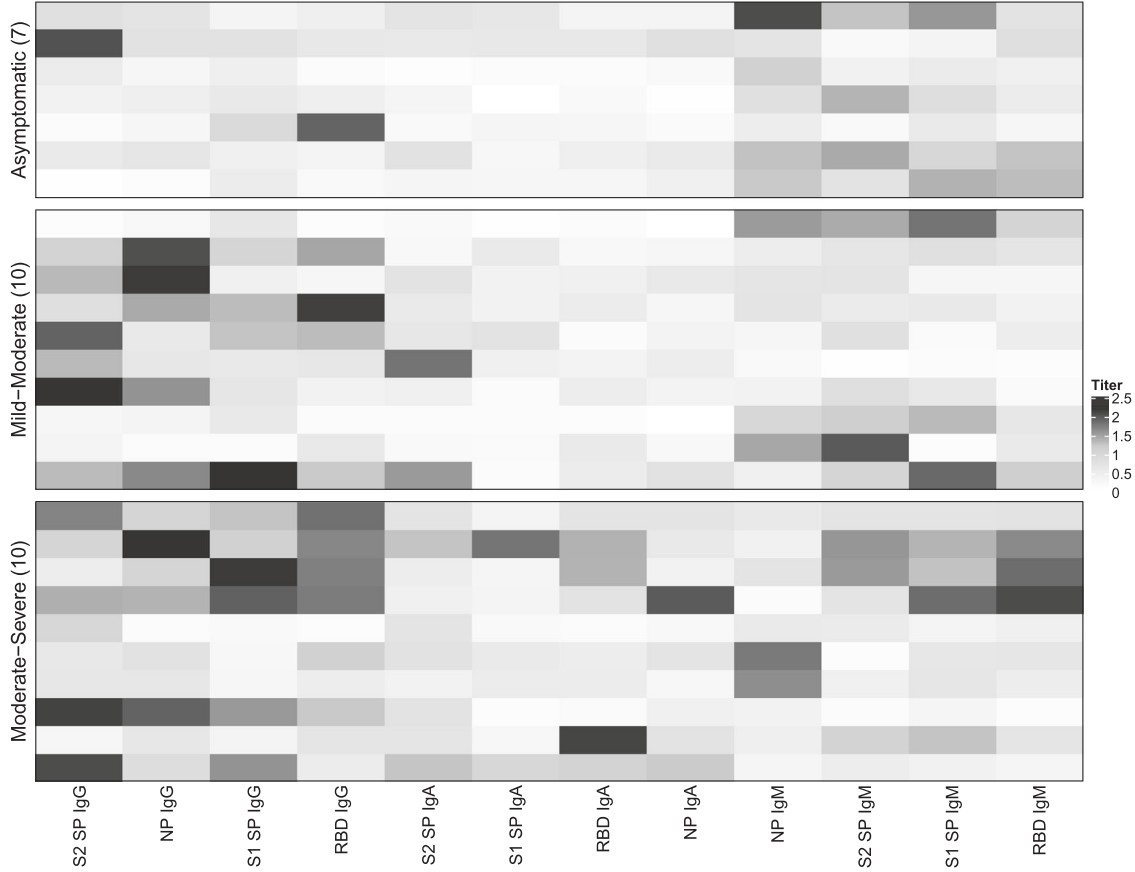

**Fig. 3 Heatmap of IgG, IgA, and IgM titers against SARS-CoV-2 antigens in asymptomatic and symptomatic COVID-19-positive participants.** Heatmap generated via an agglomerative hierarchical approach showing differences in IgG, IgA, and IgM titers against spike 1 glycoprotein (S1 SP), spike 2 glycoprotein (S2 SP), receptor-binding domain (RBD), and nucleoprotein (NP) in COVID-19-positive asymptomatic participants (n = 7), participants who experienced mild-moderate symptoms (n = 10), and participants who experienced moderate-severe symptoms (n = 10). Each row represents titers in an individual participant. The color key indicates titer, with darker shades indicating elevated titers and lighter shades indicating lower titers.

symptomatic participants than COVID-19-positive asymptomatic participants. Therefore, asymptomatic and symptomatic COVID-19-positive participants may exhibit different immune responses to SARS-CoV-2. Asymptomatic participants were more likely to test positive for IgM only. Interestingly, none of the asymptomatic participants were positive for IgA, and participants whose symptomology was classified as moderate-severe had significantly higher titer levels of RBD IgA. IgA is the principal antibody in secretions including the mucus epithelium and the intestinal and respiratory tract and acts primarily as a neutralizing antibody to eliminate pathogens before infection begins[36]. Neutralizing antibodies such as IgA typically block viral binding to surface receptors on cells and disable receptor-virus interaction. In addition, it has been reported that circulating IgA may be involved in the formation of immune complexes that amplify pro-inflammatory cytokine signaling[36]. Several recent studies have found high levels of serum RBD IgA in COVID-19 patients that are significantly correlated with symptom severity[37,38]. IgA may play a critical role in mucosal and systemic responses to SARS-CoV-2 infection and titer levels may provide an early indication of symptom severity and disease progression. Further research should be focused on characterizing the IgA response in COVID-19 patients with moderate-severe symptomology.

There are several limitations to the present study. First, the sample sizes are relatively small and the population was largely young and healthy. Participants in this study were also not uniformly subjected to PCR testing to confirm SARS-CoV-2

infection, although the chemiluminescent assay was validated using RT-PCR testing in a separate cohort of patients. In addition, serological testing was only performed at one timepoint and with varied duration from symptom onset. Future studies should include larger sample sizes, confirmation of SARS-CoV-2 infection in participants by PCR, measurements of titers at multiple timepoints with uniform duration from symptom onset, and a greater focus on a more diverse population of participants with moderate-severe COVID-19 symptomology. Nonetheless, the data presented are valuable in identifying unique COVID-19 symptomology and characterizing the immune response to SARS-CoV-2. Our findings confirm those of previous studies and provide insight regarding the immune response in symptomatic versus asymptomatic patients.

In conclusion, the data presented herein indicate that IgM antibody levels were elevated earlier in the course of symptoms in participants with a positive serological test result, and that IgG levels tended to be elevated longer than IgM. IgM appeared to decline below positive detection levels 30 days post-symptom onset. The average IgG antibody titers against all antigens were significantly greater in COVID-19-positive symptomatic participants than in asymptomatic participants. COVID-19-positive symptomatic participants exhibited a greater IgG immune response while asymptomatic participants had a greater IgM response. Our data further suggest elevated IgA levels may track with increased symptom severity. These findings could aid in distinguishing individual immune responses in a wider population for tracking purposes.

## Methods

**Assay validation cohort.** Results from a clinical sensitivity and specificity performance study performed by Vibrant American Clinical Labs on a different patient population were reported to justify using this chemiluminescent assay to diagnose participants in the current Denver, Colorado cohort as COVID-19-positive or negative. The previous clinical study included testing of a panel of retrospectively collected patient serum samples from RT-PCR confirmed SARS-CoV-2 infected ($n = 303$) and uninfected individuals ($n = 296$). In addition, serum samples from healthy controls ($n = 4,502$) and archived infectious disease controls ($n = 128$) collected prior to the COVID-19 outbreak were also included in the analyses. Disease controls included individuals with Lyme disease ($n = 20$), cytomegalovirus infection ($n = 4$), hepatitis C ($n = 20$), syphilis ($n = 6$), celiac disease ($n = 26$), systemic lupus erythematosus ($n = 26$), and rheumatoid arthritis ($n = 26$). Serological testing was performed by Vibrant America using a SARS-CoV-2 IgM, IgG, and IgA chemiluminescent assay (Vibrant COVID-19 Ab Assay; Vibrant America Clinical Labs) as described below. Clinical sensitivity and specificity at the 95% confidence levels, are displayed for overall IgM/IgG/IgA, overall IgM, IgG, and IgA, and for IgM, IgG, and IgA antibody titers against SARS-CoV-2 spike 1 glycoprotein, spike 2 glycoprotein, receptor binding domain, and nucleoprotein, and are shown in Table 1.

**Participant population- Denver, Colorado cohort.** One hundred seven non-hospitalized participants ($n = 107$) were recruited from a community clinic designated for the care of patients without severe or life-threatening symptoms of COVID-19. All study participants underwent a detailed history by a board-certified physician and filled out an online questionnaire (detailed below). Participants were excluded from the study if they were not permanent residents of Colorado or if they reported symptoms before December 25, 2019. Symptomatic and asymptomatic participants were deemed probable COVID-19 cases based on clinical presentation and/or prior known or suspected exposure to an individual with a PCR-confirmed case of COVID-19. Blood serum samples were collected from each patient as detailed below. A total of thirty-one participants ($n = 31$) had positive antibody tests, defined by a titer concentration $\geq 1$ (as defined below) for at least one antibody/antigen combination, and were deemed confirmed COVID-19 cases. Each participant in this cohort was further categorized into two groups (either "asymptomatic" or "symptomatic") based on their self-reported symptoms or lack of symptoms. This information was extracted from questionnaire responses as described below. The remaining seventy-six participants ($n = 76$) with negative antibody tests were also categorized into two groups (either "asymptomatic" or "symptomatic") based on their self-reported symptoms or lack of symptoms. Demographic information for the study population is shown in Table 2.

**Questionnaire.** Demographic information, medical histories, questions related to SARS-CoV-2 exposure and prior testing, vaccinations, travel, physical activity levels, drug use, and date of symptom onset, resolution, and severity were gathered using a 42-item electronic form that took an average of 10–15 min to complete. A total of 64 participants provided questionnaire responses. Sixty-two ($n = 62$) participants filled out the form an average of 19 days (median = 17 days; range = 3–48 days) following their initial serology antibody test. Two ($n = 2$) participants responded to the questionnaire 3 and 10 days prior to their initial serology antibody test (average = 6.5 days). The presence of specific symptoms and symptom severity were reported on a 5-point Likert scale with 0 indicating the absence of the specific symptom, 1 being a mild symptom, and 5 being a severe symptom. Reported symptoms included fever, dry cough, sore throat, fatigue, sputum production, nasal congestion, runny nose, headache, loss of smell, loss of taste, vomiting, diarrhea, dizziness, chills, body aches or myalgia, shortness of breath, swollen lymph nodes, and chest pain. A total symptom severity score was generated by adding the severity score for each of the 18 individual symptoms. The total possible symptom severity score was 90. Participants were asked to report the presence of illness between December 25, 2019 and May 26, 2020 and reported the first date they remembered experiencing symptoms and the date on which the symptoms resolved. We defined the course of illness as the period from the onset of symptoms to the date of symptom resolution.

**Blood collection.** Blood samples were obtained from all 107 participants. Each blood sample was taken between 1 and 117 days of symptom onset if the participant was experiencing symptoms. A certified phlebotomist performed each venipuncture at Resilience Code (Englewood, Colorado). From each participant and for each draw, 7.5 mL of blood was drawn from the antecubital vein and collected in one VACUETTE® serum separator tube (SST) (Greiner Bio-One, Monroe NC) containing a clot activator and gel which was provided in the chemiluminescent immunoassay kit supplied by Vibrant (Vibrant COVID-19 Ab Assay; Vibrant America Clinical Labs). Each SST tube was allowed to clot at room temperature (approximately 20 °C) for 30 min. Serum separator tubes were then centrifuged at $1500 \times g$ at room temperature for 20 min. Samples were stored at Resilience Code at room temperature for no longer than 8 h prior to shipping the same day or were refrigerated and stored at 4 °C prior to shipping to Vibrant America Clinical Labs (San Carlos, CA) the day following blood collection. Dates

and times of blood draws, duration between blood draw and shipping, and duration between blood draw and reported results were recorded.

**Serological testing.** Serological testing was performed by Vibrant America using a SARS-CoV-2 IgM, IgG, and IgA chemiluminescent assay (Vibrant COVID-19 Ab Assay; Vibrant America Clinical Labs).[26] This assay was approved by the FDA for Emergency Use Authorization (https://www.fda.gov/medical-devices/emergency-situations-medical-devices/eua-authorized-serology-test-performance; https://www.fda.gov/media/138629/download). Microbial contaminated or specimens containing visible particulate were excluded. Grossly hemolyzed or lipemic serum or specimens were avoided. The samples were stored at 2-8 °C for up to 7 days before assay. Briefly, purified recombinant SARS-CoV-2 antigens, including S1 subunit of Spike Protein (S1), Receptor Binding Domain (RBD), S2 subunit of Spike Protein (S2), and the Nucleoprotein (NP), were bound to functionalized silicon chips and assembled onto a 96-pillar plate. A layout of 8 chips on each pillar (4 chips with SARS-CoV-2 antigens and 4 reference chips used in software analysis) was created using an automated semiconductor assembly technique. The assay was performed using three 96 pillar plates for each assay (one for IgG antibody detection, one for IgA antibody detection, and one for IgM antibody detection) and an automated liquid handling workstation (Hamilton Microlab STAR). After blocking, the positive control, negative control, cut-off control, and diluted patient sera (1:50) were added to the wells and allowed to incubate for 15 min at room temperature. The plates were washed 3 times with 1X Tris-Buffered Saline containing 0.1% Tween 20 Detergent (TBST) buffer (Amresco INC, Solon OH) for 5 min each time to remove any unbound sample. A 1:2000 dilution of Goat Anti-Human IgG HRP, Goat Anti-Human IgM HRP, and Goat Anti Human IgA HRP secondary antibody was then added individually for 15 min at room temperature. After washing with TBST and DI $H_2O$, the remaining enzyme activity was measured by adding a chemiluminescent substrate (Clarity Max from Bio-Rad, Hercules CA). The intensity of the signal from each chip was measured using a high resolution chemiluminescence imager (Q-View™ Imager Pro, Quansys Biosciences, Logan UT). Each plate was scanned for 5 min. A reporter software was used to obtain the raw chemiluminescent signals.

The raw chemiluminescent signals were subjected to quantile normalization, spatial correction, and background correction. The mean ± SD of the signal intensity for each antigen was calculated from healthy controls used as the signal threshold. The raw sample results were quantitated into arbitrary chemiluminescent units by comparison with cut-off values. A sample cohort of 368 samples which consist of healthy controls collected prior to the SARS-CoV-2 outbreak was used to determine the cut-off. The upper 97th percentile was set to 1.00 for each antigen tested. Participant samples were considered to be negative for COVID-19 if the sample intensity was equal to or less than 1.00 and positive for COVID-19 if it was greater than 1.00.

**Data management and statistical analysis.** Analysis was conducted using data from a total of one hundred and seven ($n = 107$) participants. For comparison of ethnicity, BMI, smoking history (tobacco or marijuana), hypertension, cardiovascular disease, diabetes, liver disease, autoimmune disease, tick-borne illness, hemochromatosis, blood-related disease (i.e., anemia), and symptom severity, data was used from a total of sixty-four ($n = 64$) out of the one hundred and seven ($n = 107$) participants who responded to the questionnaire. Positivity (titers $\geq 1.0$) or negativity (titers $< 1.0$) for each antibody/antigen combination was known for all study participants ($n = 107$). However, antigen/antibody titer levels for five ($n = 5$) participants were missing. Therefore, plotted titers or average titers in tables and graphs include only one hundred and two ($n = 102$) participants.

Exploratory correlational analyses were performed using the total study population and within subgroups of participants who tested either positive or negative for COVID-19. Pearson's correlations were used to investigate possible relationships between demographic variables (age and BMI) and the compiled severity scores for the 18 COVID-19 symptoms that were included on the questionnaire. The data were adjusted for multiple comparisons using the Holm procedure to avoid inflation of the Type 1 error rate. Pearson's correlations were also used to generate correlograms (Supplementary Fig. 4) which display positive and negative correlations between antibody/antigen combinations in COVID-19-positive and -negative participants, along with COVID-19-positive participants experiencing no symptoms (asymptomatic), mild-moderate, and moderate-severe symptoms. Since all participants in each group were included in analysis regardless if they were positive (titers $\geq 1.0$) or negative (titers $< 1.0$) for each antibody/antigen combination, the data were not normally distributed and failed the assumption of normality required by the inferential test. Therefore, $p$-values were not reported for the correlation data.

To further investigate the relationship between symptom severity and titer concentrations, a categorical variable was generated using a median split of the total symptom severity score for positive symptomatic participants. The median symptom severity score was 21. Participants who reported a total symptom severity score equal to or below 21 were categorized as having mild-moderate symptoms while those who reported a score above 21 were categorized as having moderate-severe symptoms. Two-tailed Independent Samples T-tests were used to determine differences in levels of all 12 of the antibody/antigen combinations and the two symptom severity groups. To further understand whether certain antibody/antigen combinations might be related to mild-moderate, moderate-severe, or asymptomatic symptom presentation, a

heatmap visualization (Fig. 3) was generated using an agglomerative hierarchical clustering algorithm[39,40]. Normative titer values for each antibody/antigen were plotted in columns and participants were plotted in rows. Columns were clustered using Euclidean distance and complete linkage. One participant was removed from this analysis due to the presence of an outlier.

To test for intergroup differences in the number of participants with positive antibody/antigen combinations, certain symptoms, and demographic variables (age, BMI, gender, medical conditions, smoking history, etc.) two-tailed Independent Samples T-Tests, Chi Square Analyses or Fisher's Exact Tests were utilized. Two-tailed Independent Samples T-tests were used for continuous data. Chi Square analyses were used with categorical data that had more than 5 cases in all cells of the generated contingency tables, and Fisher's Exact Test was used with categorical data that had less than 5 cases in any one of the cells. A Chi Square Goodness of Fit Test was used to test for differences in the number of participants that tested positive for IgG or IgM antibodies with one observed variable.

General additive models (GAMs) were used to fit the data presented in Supplementary Fig. 5. To ensure the GAMs were the best fit for the data, generalized linear models (GLMs) were also generated, and an ANOVA Chi-Square Test was used to compare the two models. The GAM model fits the data significantly better than the linear model for the data presented in Supplementary Table 2 and Supplementary Fig. 5. A categorical variable was generated using the calculated days between symptom onset and the initial test to look at possible differences in titers at certain time intervals between symptom onset and the initial test. Separate one-way ANOVAs were used to test whether mean titers differed between the four intervals (0–30 days, 30–60 days, 60–90 days, 90–120 days) from symptom onset to the initial test. Tukey post hoc analyses were conducted to evaluate differences between adjusted means. All statistical analyses and data visualization were performed using RStudio for Mac (RStudio Team [2019]. RStudio: Integrated Development for R. RStudio, Inc., Boston, MA URL). The alpha level for null hypothesis rejection was set at 0.05. Data are presented as mean ± standard error, unless otherwise noted.

**Ethical approval and informed consent process**. Each patient granted Resilience Code specific, written authorization to disclose their medical records for research purposes. This study was determined to be exempt from IRB approval by the IRB at the University of Denver. All protected health information (PHI) was de-identified prior to analysis. A random global unified identifier (GUID) code was assigned to each participant and was used on all samples and forms associated with the study to maintain anonymity.

**Reporting summary**. Further information on research design is available in the Nature Research Reporting Summary linked to this article.

## Data availability
Fully de-identified data are publicly available at the following dedicated GitHub repository: https://github.com/linsemanlab/Grossberg-Koza-et-al-2021. The complete dataset used for analyses and source data for each figure is included as part of this repository. Raw data presented in Table 1 is maintained by Vibrant Clinical Labs and can be made available upon reasonable request. For inquiries, please contact Hari Krishnamurthy at hari@vibrantsci.com. Source data for raw serology and participant questionnaires, Figs. 1–3, and Supplementary Figs. 1–5 are provided as a Source Data file.

## Code availability
Custom R code supporting this publication is publicly available at the following dedicated GitHub repository: https://github.com/linsemanlab/Grossberg-Koza-et-al-2021.

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

## Acknowledgements
This project was funded by grants from the Knoebel Institute for Healthy Aging at the University of Denver to Dr. Ann-Charlotte Granholm and Dr. Daniel A. Linseman. We would also like to thank Dr. Catherine Durso (University of Denver) for her guidance in statistical analyses.

## Author contributions
D.A.L., A.C.G., C.P. supervised, and conceptualized this project. C.P. recruited participants for this study and executed clinical protocols. H.K.K. and V.J. developed the FDA Emergency Use Authorization assay, processed clinical samples, and performed SARS-CoV-2 serological testing. Research data was maintained and statistically analyzed by A.N.G. and L.A.K. The original draft was written by A.C.G., A.N.G., and L.A.K. The text was edited by A.C.G., D.A.L., A.L., A.N.G., L.A.K., with input from C.P., H.K.K., and V.J.

## Competing interests
H.K.K. and V.J. are affiliated with Vibrant America Clinical Labs, a commercial lab that performs commercial antibody testing for the novel coronavirus. All other authors declare no competing interests.
