## [Peer Review File · Nature Communications]

REVIEWER COMMENTS

Reviewer #1 (Viral immunity) (Remarks to the Author):

In this study, Grossberg et al. use a newly established automated chemiluminescent serological assay to diagnose Covid-19 infection, and to investigate the association between a positive ab response and a number of parameters that characterize the course of this infection: symptom profile, severity of symptoms, IgM and IgG antibody titer kinetics, ab specificity related to severity, and non-specific vaccination.

Major criticism:

Considering that a substantial part of this study is the validation of a new seroanalytical approach, it is very unfortunate that independent PCR testing is not applied routinely in the acute phase as a key criteria to establish a positive diagnosis, but that instead a positive serological response in any of the tested antigen/antibody combinations is used as case definition for viral infection – current or previous. Indeed, this fact is never clearly spelled out, leaving it up to the reader to figure out for him/herself that this is indeed the case.

Another critical issue is that the study as presented appears rather unfocussed, trying to cover a lot of different facets of the virus/host interaction, while at the same being quite diffuse in the presentation of key information regarding the results of the seroassay itself. In my opinion, the authors should focus on presenting the quintessential data of the Vibrant seroanalysis in a clear and legible fashion, and refrain from trying to cover a number of additional aspects, several of which have already been addressed in other studies, e.g. symptom profile in Covid-19 patients, and also the relatively rapidly declining ab levels as a function of time has now be described.

Most importantly, the information on positivity rates and actual ab levels detected in each antigen/ab combination currently presented in tables 2 and 3, could be displayed in a much more reader friendly way as a number of figures with a similar lay-out as that in fig. 1B; one for each antigen/ab combination with four groups in each: one for Covid-19 symptomatic severe-moderate, one for moderate to mild, one for asymptomatic patients and one for non-Covid-19 persons.

It is stated by the authors that The Vibrant assay is unique in its ability to detect a variety of antibody/antigen combinations, and since the data reveals that apparently some patients are not uniformly positive in all antigen/ab combinations, using an assay based on a single combination may lead to false negative results. Following this line thinking, and based on available information regarding the specificity of a number of other assays, the authors attempt to calculate the rate of false negative using results from the matching Vibrant assay. However, the reasoning behind this approach is flawed, since the sensitivity not only depends on the antigen/ab combination, but also on the applied technology. Thus, the capacity to detect low amounts of abs may be different in different types of assays, so even if no positive result is obtained using the Vibrant approach, a positive result may still be picked up, if e.g. a sensitive ELISA technique is applied instead. Unless, the authors can document equal or better sensitivity of the Vibrant approach, the only way to do an appropriate comparison is to run titrations of the same sera in the assays to be compared, and look at the concordance.

On the same note, as the data set is presented right now, it is impossible not only to judge which antigen/ab combinations are the essential to establish a positive diagnosis and which are largely irrelevant. Perhaps more importantly, does the absence of a “full house” in some patients simply reflect that these generally have responded more poorly leading to negative results for the (same) antigen/ab combinations in such patients, e.g. asymptomatics? Correlation plots depicting the association between the magnitude of the various responses in the tested individuals should be included as supplementary data. This information is essential when trying to interpret findings of ab breadth in what may be low responding groups. The question is whether there are actually qualitative differences or some antigens simply induce poorer response, and therefore these specificities are repeatedly absent, when the general responsiveness is low.

Regarding the kinetics of IgM and IgG titers as a function of time, the set-up in this study is far from optimal. To acquire the relevant information regarding magnitude, isotype switching and decline, sera harvested at different time points from a cohort of patients followed over time is required.
Minor points:

Figure legends are quite extensive, and e.g. information regarding criteria for scoring patient severe/moderate/mild is more appropriately included in Materials and Method. In that context why is the scoring described in M &M, seemingly different from the one mentioned in the legend to fig. 1? Also factual results do not belong in the figure legend, but should be presented elsewhere. The figure legends should only contain information essential to understand what is presented.

Regarding trained immunity, I believe that this study is not powered to answer that question. Generally, one would expect relatively weak effects, and to evaluate effects of this magnitude precisely, much larger sample sizes are needed. The fact that there may be a trend for flu vaccination does not eliminate my concerns, a statistical difference does not prove biological relevance, it could still be a random observation, albeit rare, or the difference could be too small to be clinically relevant

Reviewer #2 (Biostatistician, systems data analyses) (Remarks to the Author):

In this paper, the authors present the results of a novel multiplex serology test to assess the immune response to COVID-19 in an outpatient cohort consisting of adults and children in Colorado. The study is important. But I found the records presented are not accurate. Here are some comments:

- Page 6. There are 108 participants in the study. "Thirty-one participants tested positive for at least one SARS-CoV-2 antibody/antigen combination (antibody titers ≥ 1.0). Seventy-six participants tested negative for all antibody/antigen combinations (titers < 1.0). There is one subject missing. Why?

- Table 1. The records in this table are completely messed up. Ethnicity. They don't sum up to the sample size in each column.

- Table 1. The percentages reported below the Ethnicity row are no longer calculated using the sample size in each column. For example, smoking history, $10/108 \neq 15.38\%$, $4/31 \neq 14.29\%$ and $6/76 \neq 16.67\%$.

- Many numbers reported in Table 2 are also wrong. $18/31 \neq 64.9\%$, $13/76 \neq 36.11\%$.

- Table 2. Suggest displaying actual p-values to see which antibody shows the biggest difference.

- Figure 1A. Suggest using box plots to compare.

- Figure 2. Suggest using nonlinear or piece-wise linear models to fit the data. Especially for c and d, there are strong non-linear trends.

Reviewer #3 (ACE2, viral responses) (Remarks to the Author):

In this study, Allison N. Grossberg et al used a multiplex chemiluminescence immunoassay to assess the antibody response among asymptomatic and symptomatic SARS-CoV-2 infected patients.

Previous studies (such as Alberto L. Garcia-Basteiro et al published in Nature Communications; Zhang Yongchen et al published in Emerging Microbes & Infections) had already reported the IgM, IgG and IgA antibody titers against SARS-CoV-2 antigens among symptomatic and asymptomatic COVID-19 patients. The research in this study was not novel enough.

Minor comments

The antibody titers comparison between mild and severe symptoms (Figure 1) or between symptomatic and asymptomatic (Table 3) on COVID-19-positive participants were without adjusted analysis, such as age, sex and days from symptom onset.

Statistical methods were incorrect in some analysis. For example, two-tailed independent T-Tests was inappropriate used to compare categorical variables in Table3.

Antibody titers against SARS-CoV-2 antigens between males and females among COVID-19-positive and COVID-19-negative participants were not shown.

RESPONSES TO REVIEWERS' COMMENTS

Reviewer #1 (Viral immunity) (Remarks to the Author):

“In this study, Grossberg et al. use a newly established automated chemiluminescent serological assay to diagnose Covid-19 infection, and to investigate the association between a positive ab response and a number of parameters that characterize the course of this infection: symptom profile, severity of symptoms, IgM and IgG antibody titer kinetics, ab specificity related to severity, and non-specific vaccination.”

Major points:

1. Considering that a substantial part of this study is the validation of a new seroanalytical approach, it is very unfortunate that independent PCR testing is not applied routinely in the acute phase as a key criteria to establish a positive diagnosis, but that instead a positive serological response in any of the tested antigen/antibody combinations is used as case definition for viral infection – current or previous. Indeed, this fact is never clearly spelled out, leaving it up to the reader to figure out for him/herself that this is indeed the case.

RESPONSE: *The reviewer is correct that PCR testing was not performed on the cohort of participants from Colorado that were given the new serology test. However, in the revised paper we have now included data from an initial, much larger cohort showing that indeed the novel Vibrant serology assay displays greater than 99% agreement in terms of both sensitivity and specificity with PCR test results for COVID-19. These new data are shown in revised Table 1 and are presented in the first paragraph of the Results – “Initial validation of Vibrant America Clinical Labs chemiluminescence immunoassay”, page 6.*

2. Another critical issue is that the study as presented appears rather unfocused, trying to cover a lot of different facets of the virus/host interaction, while at the same being quite diffuse in the presentation of key information regarding the results of the seroassay itself. In my opinion, the authors should focus on presenting the quintessential data of the Vibrant seroanalysis in a clear and legible fashion, and refrain from trying to cover a number of additional aspects, several of which have already been addressed in other studies, e.g. symptom profile in Covid-19 patients, and also the relatively rapidly declining ab levels as a function of time has now be described.

RESPONSE: *As suggested by the reviewer, we have presented key data on validation of the Vibrant seroassay (new Table 1) and we have taken out much of the data on the symptom profile and de-emphasized to some extent the rapidly declining Ab levels as a function of time.*

3. Most importantly, the information on positivity rates and actual ab levels detected in each antigen/ab combination currently presented in tables 2 and 3, could be displayed in a much more reader friendly way as a number of figures with a similar lay-out as that in fig. 1B; one for each antigen/ab combination with four groups in each: one for Covid-19 symptomatic severe-moderate, one for moderate to mild, one for asymptomatic patients and one for non-Covid-19 persons.

RESPONSE: *This is an excellent suggestion. We have now presented the titers for all 12 antibody/antigen combinations for COVID-19-positive versus COVID-19-negative participants in new supplemental Figures S1a-c, which replaces Table 2 in the original draft (see section “Antibody titers against SARS-CoV-2 antigens” of Results, page 7). In addition, titers of IgA antibodies against RBD were statistically significantly higher in COVID-19-positive participants with moderate-severe symptoms when compared to participants with mild-moderate symptoms and asymptomatic participants (revised Figure 1a). We also found that titers of IgG against NP as well as IgA against S2 SP were significantly elevated in participants with moderate-severe symptoms compared to those*

who were asymptomatic (revised Figure 1b, c). Therefore, these data are presented in revised Figure 1 and discussed on page 9 of the Results. However, we have chosen to not show graphs of the other antigen/antibody combinations since they did not show significant differences between moderate-severe and mild-moderate symptoms or asymptomatics. Finally, because the only statistically significant differences in serology data between the COVID-19-positive symptomatic vs asymptomatic participants are in the proportion of each group that is positive for a specific antigen/antibody combination, but not the absolute titer values, we have chosen to keep these results in Table 3. Though we have revised Table 3 to be more easily read. These findings are presented in section “Antibody titers in COVID-19-positive symptomatic versus asymptomatic participants” of the Results, pages 10-11.

4. It is stated by the authors that The Vibrant assay is unique in its ability to detect a variety of antibody/antigen combinations, and since the data reveals that apparently some patients are not uniformly positive in all antigen/ab combinations, using an assay based on a single combination may lead to false negative results. Following this line thinking and based on available information regarding the specificity of a number of other assays, the authors attempt to calculate the rate of false negative using results from the matching Vibrant assay. However, the reasoning behind this approach is flawed, since the sensitivity not only depends on the antigen/ab combination, but also on the applied technology. Thus, the capacity to detect low amounts of abs may be different in different types of assays, so even if no positive result is obtained using the Vibrant approach, a positive result may still be picked up, if e.g. a sensitive ELISA technique is applied instead. Unless, the authors can document equal or better sensitivity of the Vibrant approach, the only way to do an appropriate comparison is to run titrations of the same sera in the assays to be compared and look at the concordance.

RESPONSE: *This is a valid point. We have taken out the table comparing the Vibrant assay to other serology assays and we have omitted the associated discussion.*

5. On the same note, as the data set is presented right now, it is impossible not only to judge which antigen/ab combinations are the essential to establish a positive diagnosis and which are largely irrelevant. Perhaps more importantly, does the absence of a “full house” in some patients simply reflect that these generally have responded more poorly leading to negative results for the (same) antigen/ab combinations in such patients, e.g. asymptomatics? Correlation plots depicting the association between the magnitude of the various responses in the tested individuals should be included as supplementary data. This information is essential when trying to interpret findings of ab breadth in what may be low responding groups. The question is whether there are actually qualitative differences, or some antigens simply induce poorer response, and therefore these specificities are repeatedly absent, when the general responsiveness is low.

RESPONSE: *In response to this comment, we have included correlation plots (supplemental Figures S4a, b) as well as a heat map of antigen/antibody titers grouped by symptom severity (new Figure 3) in the revised manuscript. Correlation analysis was performed for antigen/antibody combinations in COVID-19 positive and negative participants, as well as COVID-19 positive participants sub-categorized by mild-moderate symptoms, moderate-severe symptoms, or asymptomatic. The results of these analyses are presented in section “Antibody/antigen combinations in relation to COVID-19 diagnosis and symptom severity” of the Results, pages 12-13.*

6. Regarding the kinetics of IgM and IgG titers as a function of time, the set-up in this study is far from optimal. To acquire the relevant information regarding magnitude, isotype switching and decline, sera harvested at different time points harvested from a cohort of patients followed over time is required.

RESPONSE: We recognize this significant limitation in the design of our study regarding the kinetic analysis of antibody titers over time. We have documented this limitation explicitly in the Discussion on page 16. Nonetheless, our data on IgM and IgG titers decaying over time is in agreement with prior work and this is now also better explained in the revised Discussion.

Minor points:

1. Figure legends are quite extensive, and e.g. information regarding criteria for scoring patient severe/moderate/mild is more appropriately included in Materials and Method. In that context why is the scoring described in M &M, seemingly different from the one mentioned in the legend to fig. 1? Also, factual results do not belong in the figure legend, but should be presented elsewhere. The figure legends should only contain information essential to understand what is presented.

RESPONSE: The Figure legends have been modified as suggested. The scoring description for symptom severity has been moved to the M&M and is now consistent throughout the paper. Factual results have been removed from the legends.

2. Regarding trained immunity, I believe that this study is not powered to answer that question. Generally, one would expect relatively weak effects, and to evaluate effects of this magnitude precisely, much sample sizes are needed. The fact that there may be a trend for flu vaccination does not eliminate my concerns, a statistical difference does not prove biological relevance, it could still be a random observation, albeit rare, or the difference could be too small to be clinically relevant.

RESPONSE: We have removed the data and discussion regarding the effects of flu vaccination.

Reviewer #2 (Biostatistician, systems data analyses) (Remarks to the Author):

“In this paper, the authors present the results of a novel multiplex serology test to assess the immune response to COVID-19 in an outpatient cohort consisting of adults and children in Colorado. The study is important. But I found the records presented are not accurate. Here are some comments:”

1. Page 6. There are 108 participants in the study. “Thirty-one participants tested positive for at least one SARS-CoV-2 antibody/antigen combination (antibody titers ≥ 1.0). Seventy-six participants tested negative for all antibody/antigen combinations (titers < 1.0).” There is one subject missing. Why?

RESPONSE: One participant in the original study filled out the symptom questionnaire but did not have serology performed. This participant has been removed from the study.

2. Table 1. The records appear in this table is completely messed up. Ethnicity. They don't sum up to the sample size in each column. Also Table 1. The percentages reported below the Ethnicity row are no longer calculated using the sample size in each column. For example, smoking history, $10/108 \neq 15.38\%$, $4/31 \neq 14.29\%$ and $6/76 \neq 16.67\%$.

RESPONSE: The demographic information is now shown in Table 2. Many of the calculations are only based on the participants who completed the health survey ($n=64$), not the total study population ($n=107$). This accounts for the discrepancies in the percentages noted by the reviewer. The Table has now been shaded to differentiate the number of participants who completed the health survey and everything in the shaded area reflects this n value. We hope this alteration will be sufficient to clarify the data shown in Table 2.

3. Many numbers reported in Table 2 are also wrong. 18/31 != 64.9%, 13/76 != 36.11%.

RESPONSE: *This is now Table 3. We apologize for these discrepancies, and the calculations have been corrected.*

4. Table 2. Suggest display actual p-values to see which antibody show the biggest difference.

RESPONSE: *This is now Table 3. Actual p-values have been included.*

5. Figure 1A. suggest using box plots to compare.

RESPONSE: *We agree with the reviewer, and Figure 1 has been revised accordingly.*

6. Figure 2. Suggest to use nonlinear or piece-wise linear models to fit the data. Especially for c and d, there are strong non-linear trends.

RESPONSE: *We have modified Figure 2 as suggested by the reviewer. We used general additive models (GAMs) to fit the data presented in Figure 2. To ensure the GAMs were the best fit for the data, generalized linear models (GLMs) were also generated, and an ANOVA Chi-Square Test was used to compare the two models. The GAM model fit the data significantly better than the linear model for the data presented in Figure 2a, 2c and 2d (see new supplemental Figure S5 and supplemental Table S2). These changes are described in the Methods on page 24.*

Reviewer #3 (ACE2, viral responses) (Remarks to the Author):

“In this study, Allison N. Grossberg et al used a multiplex chemiluminescence immunoassay to assess the antibody response among asymptomatic and symptomatic SARS-CoV-2 infected patients.”

1. Previous studies (such as Alberto L. Garcia-Basteiro et al published in Nature Communications; Zhang Yongchen et al published in Emerging Microbes & Infections) had already reported the IgM, IgG and IgA antibody titers against SARS-CoV-2 antigens among symptomatic and asymptomatic COVID-19 patients. The research in this study was not novel enough.

RESPONSE: *We appreciate the reviewer’s comment. The Garcia-Basteiro paper focused on a large population of healthcare workers in Spain (578 participants of which 54 were seropositive). The Yongchen paper focused on a rather small cohort of 21 COVID-19-positive patients in the Jiangsu Province of China (11 non-severe, 5 severe, and 5 asymptomatic carriers). To extend these previous studies, we report the use of a novel seroanalytical tool, the Vibrant serology test, which examines seropositivity for IgA, IgG, and IgM antibodies against 4 distinct SARS-CoV-2 antigens, spike S1, spike S2, receptor binding domain, and nucleocapsid – contrary to the recently published manuscripts. Thus, our study represents a more comprehensive sero-analysis of antigen/antibody combinations in COVID-19-positive symptomatic and asymptomatic patients than what was presented in these two prior studies.*

Minor points:

1. The antibody titer comparison between mild and severe symptoms (Figure 1) or between symptomatic and asymptomatic (Table 3) on COVID-19-positive participants were without adjusted analysis, such as age, sex and days from symptom onset.

RESPONSE: We did not find significant differences in titers of IgA antibodies against RBD, IgA against S2 SP, or IgG against NP between COVID-19-positive female and male participants in these symptom groups (see supplemental Figure S3 and highlighted paragraph on page 9 of Results). There were also no significant differences in days between symptom onset to serological test date, age, sex, and BMI between COVID-19-positive participants with mild-moderate versus moderate-severe symptomology (data not shown). Regarding COVID-19-positive symptomatic and asymptomatic participants, there were no significant differences in sex, age, BMI, smoking history, or the presence of chronic illnesses between these two groups when analyzed by two-tailed Independent Samples T-Tests (Table 3 and data not shown; see page 10 of Results). Therefore, adjustments for these parameters was not deemed necessary, as now better described in the revised manuscript.

2. Statistical methods were incorrect in some analysis. For example, two-tailed independent T-Test was inappropriate used to compare categorical variables in Table 3.

RESPONSE: For the statistical analyses in Table 3, *p* values were determined via two-tailed Independent Samples T-Tests for continuous data, Chi-Squared Tests for categorical data with >5 cases per cell and Fisher's Exact Tests for categorical data with <5 cases per cell. This is now better described in the revised methods.

3. Antibody titers against SARS-CoV-2 antigens between males and females among COVID-19-positive and COVID-19-negative participants were not shown.

RESPONSE: The antibody titers for males and females among COVID-19-positive and –negative participants are now shown in supplemental Figure S2a-c and are described in the Results on page 7. Titers for all 12 antibody/antigen combinations were significantly increased in COVID-19-positive versus COVID-19-negative participants (supplemental Figure S1a-c). However, titers were not significantly different between males and females (supplemental Figure S2a-c). This is now better described in the revised manuscript.

REVIEWERS' COMMENTS

Reviewer #1 (Remarks to the Author):

The authors have for the main part responded adequately to my comments. Thus, in my opinion the data presentation is appropriate now. My remaining concern is related to the novelty of the presented observations and the extent to which these findings will impact our understanding of the Covid-19 disease

Reviewer #2 (Remarks to the Author):

the authors have addressed all my comments.

Responses to Reviewers for Nature Communications manuscript NCOMMS-20-26622A

Title: A multiplex chemiluminescent immunoassay for serological profiling of COVID-19-positive symptomatic and asymptomatic patients

First and corresponding authors: Allison N. Grossberg and Daniel A. Linseman

Reviewer #1 (Remarks to the Author):

The authors have for the main part responded adequately to my comments. Thus, in my opinion the data presentation is appropriate now. My remaining concern is related to the novelty of the presented observations and the extent to which these findings will impact our understanding of the Covid-19 disease.

RESPONSE: We appreciate the Reviewer's comments and we have summarized the major observations of our study below along with a statement of how we believe these findings may impact our understanding of COVID-19 disease in the future.

In the current study, we analyze symptomatology and antibody profiles of otherwise healthy, community-dwelling participants who reported a variety of COVID-like symptoms to an outpatient clinic in the Denver, Colorado area. Using the Vibrant multiplex chemiluminescent immunoassay, we find that the average IgG antibody titers against all SARS-CoV-2 antigens are significantly greater in COVID-19-positive symptomatic participants than in asymptomatic participants. COVID-19-positive symptomatic participants exhibit a greater IgG immune response while asymptomatic participants show a greater IgM response. Finally, elevated IgA levels track with increased symptom severity. These findings suggest that the Vibrant immunoassay may be useful in differentiating individual immune responses that are reflective of distinct symptom severities in those infected with SARS-CoV-2.

Reviewer #2 (Remarks to the Author):

The authors have addressed all my comments.

RESPONSE: We appreciate the Reviewer's comments.